# Crisis Management Experience from Social Media: Public Response to the Safety Crisis of Imported Aquatic Products in China during the Pandemic

**DOI:** 10.3390/foods12051033

**Published:** 2023-02-28

**Authors:** Ru Liu, Min Liu, Yufeng Li, Linhai Wu

**Affiliations:** 1School of Economics and Management, Shanghai Ocean University, Shanghai 201306, China; 2School of Humanities and Management, Guilin Medical University, Guilin 541199, China; 3School of Business, Jiangnan University, Wuxi 214122, China

**Keywords:** imported food safety, crisis management, online public opinion, text mining

## Abstract

China’s outbreak related to cold-chain aquatic product quality and safety in 2020 caused public panic and further led to a crisis in China’s aquatic industry. This paper uses topic clustering and emotion analysis methods to text-mine the comments of netizens on Sina Weibo to study the main features of the public’s views on the administration’s crisis management measures and to provide experience for future imported food safety management. The findings show that for the imported food safety incident and the risk of virus infection, the public response had four types of characteristics: a higher proportion of negative emotion; a wider range of information demand; attention paid to the whole imported food industry chain; and a differentiated attitude towards control policies. Based on the online public response, countermeasures to further improve the management ability of imported food safety crises are proposed as follows: the government should pay active attention to the development trend of online public opinion; work more on exploring the content of public concern and emotion; strengthen the risk assessment of imported food and establish the classification and management measures of imported food safety events; construct the imported food safety traceability system; build a special recall mechanism for imported food safety; and improve the cooperation between government and media, enhancing the public’s trust in policies.

## 1. Introduction

Food safety is not just reflected in daily supervision—crisis management during emergencies is also an important guarantee to ensure food safety [1]. In the context of global economic development, China’s trade volume of imported food is increasing, but at the same time, the contradiction between imported food supply and food safety is becoming increasingly prominent [2]. The demand for public health protection is increasing continuously. How to improve the crisis management ability of imported food safety incidents to meet the needs of the public has become the key topic of current food safety management.

Crisis events usually evolve rapidly with incomplete information, high uncertainty, and strong time urgency [3,4]. In the face of crisis events, managers need to make key decisions and take effective measures to eliminate threats. Due to the high frequency of food safety [5] and food rumor [6,7] in recent years, scholars have conducted a lot of research on food safety crisis management [7,8,9,10,11], finding that the interaction between the government and the public is an important part of crisis management, and social media is an important media for information dissemination in crisis events. In the context of social media, information dissemination changes from one-way government dominance to multi-way discourse interaction. The public expresses their views and participates in crisis management via social media; the government sectors participate in public opinion events and respond to the public via social media. Two-way communication between the government and the public has become the new norm for food safety crisis management in China.

However, most of the studies in the current literature focus on domestic food safety incidents with little research on imported food safety events, and there is a lack of research on imported food safety events with the risk of virus infection. At the same time, most of the literature on food safety crisis management focuses on the research on crisis management strategies and the implementation effects adopted by the government and enterprises [1,12,13,14,15,16,17]. However, insufficient attention is paid to the research on public attitudes toward food safety crisis management. The public’s online participation during the crisis has greatly increased the production and dissemination of information [18], but most of the public does not have practical experience in dealing with crises, mainly learning from influential social media users or other sources [19]. Public participation in online discussions during the crisis is mainly to express negative emotions, and people tend to overreact to food safety issues due to a low trust in supervision [20]. To increase the public’s confidence in food safety, it is necessary to systematically understand the public responses to and expectations of food safety crisis management.

Most of the current food safety crisis management studies adopt the methods of in-depth interviews [15,17] and case studies [18,19,21,22]. For example, Song et al. (2020) established a case database of food safety incidents and constructed a matching model to locate similar historical cases and help regulatory authorities make emergency responses. However, most of these research methods are aimed at crisis management institutions and enterprise staff, lacking public evaluation of food safety crisis management. Some scholars have quantitatively studied consumers’ cognition of food safety events [23] and the evaluation of crisis management [24]. However, there are still limitations, including lack of sufficient samples and a real environment; thus, these studies cannot truly reflect the public’s views during actual food safety events.

Online public opinion has the characteristics of wide sources and rich information, which can easily and quickly collect public opinion [25]. At present, the method of using online public opinion to study food safety issues is gradually becoming popular around the world. The mainstream research methods mainly use machine learning, natural language processing and other technologies for public opinion topic extraction [26] and sentiment analysis [27]. The content mainly focuses on public opinion warning research [28], public opinion evolution and communication research [29], and public opinion governance research [30]. However, there are few studies on food safety crisis management through online public opinion.

During the pandemic, China suffered an outbreak affecting imported aquatic product safety, and it successively took restrictive measures on aquatic product imports from various countries. This paper took the 2020 Chinese imported cold-chain aquatic product safety incidents as examples and analyzed the public comments on Sina Weibo to study the public’s attitude toward the government’s management of this crisis. The rest of this paper is organized as follows: Part 2 introduces the research subject, data, and research methods; Part 3 summarizes the characteristics of online public responses to crisis management; Part 4 proposes strategies to enhance the government’s ability to manage imported food crises; Part 5 discusses what makes this paper different from other similar studies and points out the limitations and future research directions; and Part 6 concludes the research results.

## 2. Materials and Methods

### 2.1. Research Subject and Data

On 13 June 2020, the incident wherein a novel coronavirus was detected from the cutting board of imported salmon attracted considerable attention from the media and netizens. Subsequently, the existence of a novel coronavirus was also detected on the outer packaging and the sample surface when the customs officers inspected the imported cold-chain aquatic products. The public expressed anxiety about whether imported aquatic products or even domestic aquatic products could be consumed.

Compared with other food safety incidents, this incident had three aspects of particularity: (i) the safety risk of imported food; (ii) the infection risk of COVID-19; (iii) the long duration. We conducted a retrospective observational study on the safety events of imported aquatic products in 2020 on Sina Weibo. It is public domain, enabling users to share text, images, videos, and other content on Weibo as well as comment on and retweet content posted by other users. According to its press release from December 2020, it has more than 521 million active users [31] and has become the most popular social platform of its type in China [32]. The Sina Weibo was the main platform for information spreading on food safety events during the epidemic. This paper takes the content of blogs about imported cold-chain aquatic products published by the users of Sina Weibo and the comments of netizens as the data source, and analyzes the topics of netizens’ attention and emotions.

#### 2.1.1. Data Acquisition and Preprocessing

In this article, public opinion data on imported cold-chain aquatic products during the outbreak of COVID-19 were collected through Python software [33,34,35], with “COVID-19”, “pandemic”, and “imported aquatic products” as the keywords. The period was set from 1 June 2020 to 31 December 2020 because the amount of Weibo-related content after 31 December 2020 was greatly reduced, and follow-up data collection stopped. The collected content included Weibo ID, text, comments, release time, number of shares, number of comments, number of likes, user ID, user nickname, etc. In total, 7131 microblog texts and 13,818 netizen comments were collected. Through data cleaning, the content in texts and comments that was irrelevant or meaningless to the public opinion of imported cold-chain aquatic products was then eliminated, and finally, 7128 valid blog posts and 7631 netizen comments were obtained.

#### 2.1.2. Online Public Opinion Stage Division Based on the Four-Stage Model

Life cycle theory has been widely used in many fields, referring to the process from the beginning to the end of an incident [36]. Currently, scholars have put forward a variety of division approaches to network the public opinion life cycle. For example, the six-stage public opinion communication model of preparation, outbreak, spread, repeat, fade and long tail for the online public opinion regarding public health emergencies is constructed based on the life cycle stage theory in crisis management [37], and the four-stage communication model of outburst, outbreak, cooling and out-of-focus is structured based on the four-point four-stage theory [38]. The life cycle theory is also applicable to the development of online public opinion regarding imported food safety as an emergency. This paper conducts a time-series analysis of the online public opinion data on the quality and safety of imported cold-chain aquatic products during the pandemic and calculates the heat degree of special events at different stages based on the numbers of microblogs, likes, shares and comments [39].

The evolution of the number of posts and comments related to online public opinion is shown in Figure 1. It can be observed that there were very little data before 12 June 2020, reaching the first small peak on 15 June. Then, the number of posts and comments began to decline, showing a slight fluctuation. Four peaks occurred from mid to late June to late October, when the collective public opinion was in the spreading stage. The maximum peak stage of public opinion took place on 13 August, reaching its highest in the whole stage. Later, the number of posts and comments gradually reduced from November to December, indicating the end of the public opinion event. This series of developments was consistent with the four-stage model of online public opinion communication.

To more accurately divide the public opinion event stages, the paper identifies the most influential microblogs on Sina Weibo based on the heat value.

On 12 June 2020, the Fengtai District of Beijing reported confirmed cases of COVID-19. This signaled the re-emergence of COVID-19 confirmed cases after 56 consecutive days of no new cases in Beijing, and the highest popularity value of related microblogs reached 1855.97. People are generally worried about the cause of COVID-19 infection and are eager to know about patients’ action track and the possible source of infection.

On 13 June 2020, the market chairman pointed out that COVID-19 was detected from the cutting board of imported salmon when sampled by the relevant departments. After the news was released, the number of discussions by Weibo users on imported salmon and COVID-19 increased rapidly. Fan and other experts said that the salmon contamination did not represent COVID-19 infections (with a heat value of 4761.805), and most netizens asked experts how to clean raw products. The journalists of China Business Network called customs and sought confirmation as to why the quarantine of imported aquatic products did not include COVID-19 detection. Hence, netizens’ negative emotions toward supervision omissions began to spread, and since then, the quantity of related Weibo posts increased significantly. On 14 June, salmon were taken out of the market everywhere. From 15 to 17 June, more provinces began to carry out nucleic acid testing of fresh, imported aquatic products, agricultural wholesale market staff, and the environment.

Customs announced on 18 June that the sampling test results of COVID-19 risk for imported goods were all negative, but consumers’ concerns about imported frozen aquatic products continued, and the salmon crisis further triggered the crisis of the aquatic industry. From 22 to 25 July 2020, CCTV News and other media released the news that business staff related to imported aquatic products in Liaoning and Dalian were diagnosed positive. This once again triggered a discussion among netizens with the heat value as high as 12,553.605, and netizens’ panic about imported aquatic products escalated. On 13 August 2020, the news that positive results of COVID-19 were detected on the surface of imported frozen meat products broke out. Netizens’ cognition about the spread of COVID-19 and imported aquatic products was overturned, causing heated discussion again.

After 27 October 2020, many media authorities released the research conclusion that the COVID-19 virus in Beijing’s Xinfadi came from imported cold-chain food. On November 9, the joint prevention and control mechanism comprehensive group of the State Council issued the “Work Plan for Preventive Comprehensive Disinfection of Imported Cold Chain Food”. The mystery of how the novel coronavirus spread was gradually unraveled, the number of related microblogs began to decrease, and the heat of the event also gradually declined.

According to the contents shown in Figure 1 and Table 1, this paper sets the specific dates of the four stages of public opinion, in which 12 June 2020 is Stage I, which is the incubation stage of the event; 13 to 17 June is Stage II, which is the outburst stage of the event; 18 June to 27 October is Stage III, which is the spread stage of the event; and 27 October to 31 December is Stage IV, which is the end stage of the event.

### 2.2. Research Method

#### 2.2.1. Topic Mining of Different Public Opinion Stages

This paper uses Chinese natural language processing technology for the text mining of public opinion. First, text comments were divided into words and stop words were removed. Before the word segmentation, a special dictionary was constructed, involving specific terms related to the food safety incident and the pandemic, such as “novel coronavirus, source of infection, normalization, Xinfadi, seafood market”, etc. Then, the stop word list of the Harbin Institute of Technology [40] was used to delete the content that did not affect the original meanings of the netizens’ comments, such as special symbols, connectives, modal words and other content, and the Jieba library [41] was used to classify the comment text.

Second, keywords were extracted to calculate the weight of words in the comment text. TF-IDF (Term Frequency-Inverse Document Frequency) algorithm is suitable for filtering words with high frequency and low discrimination, retaining important words with low frequency and high discrimination [42]. Therefore, this paper adopted “word frequency-inverse document frequency”, which is the text word weighting method based on statistics, to evaluate the importance of words to documents [43]. TF-IDF is the product of TF and IDF, where TF represents the word frequency of each word after the segmentation in the documents, and IDF represents the word category discrimination ability. The greater the weight obtained by TF-IDF calculation, the more important the word—see Formula (1).
(1)$${\mathrm{TFIDF}}_{w}=\frac{n_{w}}{N}\times \log\left(\frac{M}{m_{w}+1}\right)$$
where $n_{w}$ is the frequency of words w in the text set, N is the total number of words in the text set, $m_{w}$ is the text data of words w in the text set, M is the text and the text data contained, and ${\mathrm{TFIDF}}_{w}$ is the word frequency-inverse document frequency of words w.

Afterwards, the netizen comment text based on TF-IDF calculation was imported into the LDA theme model [44] to topic mine the whole corpus of netizen comments. The topic distribution of the netizen comments was obtained by calculating the distribution probability of the words in the comments. In this paper, the web-based interactive LDAvis tools developed by Sievert and Shirley [45] were selected to visualize the content of the topic mining section. By setting the number of topics, the topic clustering was visualized on the interaction page. The results of LDAvis visualization could answer the meaning of each topic, the attention of each topic, and the relationship between each topic. The visualization panel contains two basic parts. The left panel shows a global view of the theme model, showing the relationship between each theme and its attention. The bars in the right panel represent the top 30 words with the highest frequency from the currently selected topics, which can be used to help determine the meaning of the topic. The LDAvis system changes the correlation of words and topics by adjusting the parameter λ. When λ approaches 1, it indicates that the words appear more frequently under a topic and are more related to the topic. When λ approaches 0, the words selected are more differentiated from other topics. In this paper, adjustment was attempted in the LDAvis interface to choose more appropriate topic words.

According to the visualization results of the LDAvis topic clustering, the results that topics over-intersected and that topic clustering were unclear were eliminated to eventually determine 9 topic categories of netizen comment content at the incubation stage, 12 topic categories of netizen comment content at the outburst stage, 16 topic categories of netizen comment content at the spread stage, and 5 topic categories of netizen comment content at the end stage ending (Figure 2). High-frequency words of each topic were chosen for content summary (Table 2).

#### 2.2.2. Emotion Analysis at Different Stages of Public Opinion

For emotion analysis, this paper adopts the emotional tendency analysis function of natural language processing technology in Baidu AI Cloud [46]. Based on deep learning training, it has a high generalization ability with higher overall accuracy relative to other emotional tendency analysis methods. It can still maintain high effects in relatively long sentences, which is therefore suitable for emotional judgment in Weibo comments. The main analysis step was to call the API of Baidu natural language processing emotion tendency in the Python environment to determine the emotional polarity and to calculate the emotional value of the netizen comments during the public opinion cycle. The [0, 0.5) section represents the negative emotion value, and the (0.5, 1] section represents the positive emotion value, As shown in Table 3.

## 3. Online Public Response Based on the Four-Stage Public Opinion

Online public opinion is the public’s social position and emotional expression caused by emergencies, which can comprehensively reflect the existing social contradictions. Through the topic mining and emotional analysis of the content that netizens focused on in the four stages of imported cold-chain food during the epidemic period, it can be found that for imported food with a risk of virus infection, the public response had four types of characteristics.

First, there is a higher proportion of negative public sentiment. Compared with other online public opinion events, negative emotions account for about 30% [47], and negative comments in food safety events account for about 50% [26]. In this event, the overall negative comments of public opinion accounted for 67%, and the overall negative emotions of netizens were at a high level. During stage I, the number of netizen comments was small, with negative comments accounting for 64%, and the attention content was mainly newly increased cases. During stage II, the negative emotions of the netizens were high, and the emotional mean of the negative comments is the lowest (0.066, 2) of the four stages, and the majority of the netizens criticized the concerning content. When Customs announced the testing of imported cold-chain aquatic products and frozen products, stricter epidemic prevention and control measures were adopted everywhere. Although the netizens remained concerned about the safety of imported aquatic products, their emotions were relatively moderate, and the emotional mean of the positive comments reached the highest (0.868, 0) of the four stages. At the end of public opinion, the imported food safety incident was concluded, but the proportion of negative comments reached the highest (72%) in the four stages. Negative emotions were still spreading, mainly because the public was concerned about the spread of the pandemic abroad and the safety of imported cold-chain aquatic products.

Second, there was a wider range of public demand for information. The public not only wanted information about food safety, but also about the status of the outbreak spread by food distribution and the symptoms of the patients. As shown in Table 2, since the sudden outbreak of the salmon incident in Stage I, the trail of patients’ movements and symptoms were always the main focus of the public during Stages I and II (Topic 1–6\1–7\1–8\2–2). In addition, during Stage II and Stage III, the proper food processing methods became a major topic of public concern so as to avoid the possible safety risks of the purchased food materials (Topic 2–5\3–4).

Third, the public’s attention expanded from a single food incident to the whole imported food industry chain. As shown in Table 2, netizens questioned the food safety management during Stage II. Netizens believed that Customs lacked COVID-19 detection in the food import testing process (Topic 2–6). The questions regarding the whereabouts of the salmon and fears that salmon carried the virus reflect the netizens’ concern regarding traceability in food safety incidents, product recall and food safety supervision (Topic 2–3\2–12). Moreover, the public noted that in the food sales process, salespeople did not wear masks as required. Stage II was the initial outbreak of imported aquatic safety incidents. Out of fear, the public was more critical of food safety management, and regulatory omission caused more obvious negative emotions. During Stages III and IV, the public began to suspect the source of the virus. Speculation of virus contamination (Topic 3–2\3–10) and support for import suspension (Topic 3–9\4–2) demonstrated the public’s negative opinion of the safety of foreign food producers. Meanwhile, the public believed that domestic food processing may also be a way to cause the spread of the virus through food. Stage III was the spreading period of the incident. More and more imported aquatic products and their outer packaging were found to be carrying the coronavirus, and public trust of the safety of imported aquatic products was further reduced.

Fourth, for imported food safety incidents with a risk of virus infection, the public’s attitude towards control policies was differentiated, with low trust in the government. The public’s attitude towards the rationality of management measures for food safety incidents was divided. In the absence of an investigation, on the one hand, the public wanted the government to ban the import of aquatic products (Topics 3–9\4–2), and on the other hand, they were dissatisfied with the local government’s “one-size-fits-all” policy to ban the breeding of bamboo rats in order to reduce the risk of spreading the epidemic (Topic 3–7). In addition, the public had a low amount of trust in the government and experts, and they suspected that the government might have controlled the comments (Topic 1–5); they expressed their dissatisfaction with the efficiency of government information releasing during both Stages I and II (Topic 1–9\2–1). They also considered the views stated by the experts to be vague and difficult to understand (Topic 2–8\3–12). In Stage III, after frozen chicken wings tested positive for COVID-19, the public realized that imported cold-chain aquatic products may not be the only ones to be infected. They were disappointed by the omission of imported food safety supervision (Topic 3–11\3–16). Some consumers choose to stop buying or eating all products involved, including not only imported aquatic products but also domestic aquatic products, to avoid the risk to their health. These topics show that regulatory omissions can accelerate the destruction of public trust in the government in the face of imported food incidents that increase the risk of infection, and force some people to take more extreme protective measures.

## 4. Improving the Imported Food Safety Crisis Management Ability Based on Online Public Response

Based on the public’s response to the safety incidents concerning imported food during the epidemic period, this paper proposes four countermeasures to improve the crisis management ability of imported food organizations: (i) with social media becoming the main avenue for the public to obtain and spread information, the regulatory departments should pay attention to public reactions to food safety incidents and take corresponding measures accordingly to the stage of public opinion development; (ii) they should strengthen the risk assessment of imported food safety and establish a classification management approach for imported food safety incidents; (iii) they should build a safety traceability system for imported food and establish a special recall mechanism for the safety of imported food; (iv) they should strengthen the cooperation and division of labor between the government and the media, and focus on the public’s demands for information on multiple aspects of food safety emergencies.

Modern crisis management research has found that unified, accurate and timely information release is not only conducive to reflecting the credibility of the government, but is also better for meeting the requirements for the public’s right to know, to participate, to express and to supervise [48]. During Stage I, the public was vulnerable to the interference of “hearsay information” due to the concern about the spread of the epidemic and the inability to obtain sufficient information, with an obvious emotional fluctuation. At this time, the relevant government departments should try to release the information they have mastered, and they should curb the spread of rumors from the source in a fast and frank manner to appease public emotion. Public opinion in Stage II is the defining moment for the development of the event, and the voice of relevant personnel determines the direction of the event, as it was with the salmon incident in Xinfadi in Beijing. At this stage, the public opinion supervision department should identify the truth, guide authoritative experts to popularize science in time, and avoid the generation and dissemination of “doubtful facts”. During Stage III, the event kept developing and changing, and an increasing number of imported cold-chain products tested positive for COVID-19. At this time, while announcing the test results, the authority should have paid attention to responding to the public’s emotional needs and reducing the public’s concerns about food safety.

There are diverse categories of imported food information, involving food types, years, places of origin, manufacturers, processors, sellers and other information. During the epidemic period, the sources of imported food require different prevention and control policies. In order to avoid the risk of epidemic importation via food from abroad, a big data-based data collection platform for imported food risk classification should be established to timely assess the safety risks of each country and ensure imported food safety. At the same time, in the crisis management of food safety events, it is suggested to distinguish the categories of food safety incidents and choose different management measures according to infectious or non-infectious features. For imported food safety events with a risk of infection, a special team should be formed to monitor and improve the efficiency of the processing and feedback of the food so that prompt measures can be taken for food imports from different countries.

Food traceability and recall are important links to ensure the proper treatment of food safety incidents. At present, China lacks a well-established food traceability system [49], and the regulation of imported food is sectional. Customs is responsible for the import process, while the market circulation process is supervised by the market supervision and management department. Coordination and cooperation among different departments are challenges during crisis management [50]. Additionally, the import and sales information of food is documented by the importer, and there is a risk of false or omitted records [51]. The Measures for Food Recall Management [52] specifies that the responsible entity for the recall of imported food in China is the food importer, but few enterprises initiate recall, and the mandatory recall by the China government is currently the main way to deal with imported food problems [53]. As a result, by establishing an imported food traceability system, information on the circulation of imported food is collected to ensure that all the links of imported food from production to consumption can be traced. Thus, when there is an imported food quality and safety incident, the problem food can be quickly located. Moreover, the government should build a special recall mechanism for imported food, clarify the responsibilities and obligations of food importers, and ensure the timely recall of problem foods to reduce the impact on the public.

The government should improve the information disclosure mechanism to ensure the timeliness of information release through social media; it should give full play to the functional orientation of various government media, use official media accounts such as CCTV News to release national and local policies. The authoritative official media is characterized by high public trust and can stabilize public confidence during crisis management. In addition, mainstream media such as People’s Daily can be used to propagate the content of popular science and play a complementary role in the content of official media, meeting the public’s information needs after the outbreak of imported food events with a risk of virus infection. When facing imported food safety events with a virus infection risk, the public is more likely to display panic, and meeting the public’s multifaceted information needs promptly can help the public treat the problem with a positive attitude, thus avoiding the economic losses of more related food enterprises.

## 5. Discussion

Previous food safety events have tended to be of short duration and have allowed for the rapid identification of the cause of the incident [54]. By contrast, in this study, the cycle of imported aquatic product safety events lasted up to six months and was accompanied by multiple shifts. As a result, prolonged incident reporting led to a greater tendency for the public to appear negative and to spread their attention to multiple links involving food imports. In addition, previous infectious food safety incidents were mostly due to the food itself carrying viruses or pathogenic bacteria [55,56,57,58,59], while the risk of this incident mainly resulted in the possibility of food being contaminated with viruses. States adopted different policies to prevent and control the epidemic [60], making it impossible to guarantee the safety level of imported food. At the same time, the rapid spread of the coronavirus and the uncertainty of the disease increased public fear. Therefore, the omission of food safety regulation will undermine public trust in the government more quickly, leading to questions and criticism.

Our study had several limitations. This was a study of a collection of public views from social platforms, and the selection of media platforms is its major weakness. Despite the overwhelming use of Weibo, a large number of people still prefer to use other social platforms or will not use them, and they may have different views on events. In addition, regarding data collection, we only gathered textual data, while other data such as pictures and videos that exist on social media platforms were not analyzed, and we merely analyzed the crisis management from the public’s perspective, lacking information on the government’s side. Moreover, in the section on emotion analysis, we did not conduct a detailed classification of the emotions of the public comments. For crisis events, the public often shows fear, anger, worry and sympathy [61], and a classified treatment of emotions can better facilitate our understanding of public attitudes toward such food safety events so that we can propose more targeted management measures.

In future research, we may attempt to obtain data from multiple social platforms and expand the form of data into images, videos and other content. In addition, the emotions of online public opinion can be subdivided to investigate the factors that influence public emotions.

In conclusion, we demonstrated that for food safety events with comparatively long duration and frequent event shifts, the results of Internet opinion data mining can give us a detailed understanding of the public’s reaction to crisis management. For food safety events with a risk of virus transmission, the public’s emotional and informational needs are greater and deserve attention. In addition, information collection and risk assessment are essential to securing the safety of food under the risk of epidemic or disease transmission, as the safety degree of imported food varies from country to country. Our findings will be applicable to other regions of the pandemic and are equally applicable to possible future imported food safety events with a risk of virus transmission.

## 6. Conclusions

This paper explored the public’s reaction to the management of this imported food safety crisis from the perspective of online public opinion. The completion of the study indicates that there are diverse information needs among the public during food safety events with viral transmission risks, and negative emotions exceed sixty percent in all stages of public opinion. In addition, the safety of imported foods and management measures have not been given sufficient attention in previous efforts. Information on imported food includes both overseas production and domestic distribution, and government departments are limited in their ability to supervise overseas producers. The safety of the source of imported aquatic products and the negligence of Chinese customs officers in testing led to this incident. Public sentiment is heightened in the face of food safety events that can transmit disease through imported food. Therefore, the government should respond to the public’s informational and emotional needs in a timely manner and bridge the gaps that exist in the current management of imported food safety.

## Figures and Tables

**Figure 1 foods-12-01033-f001:**
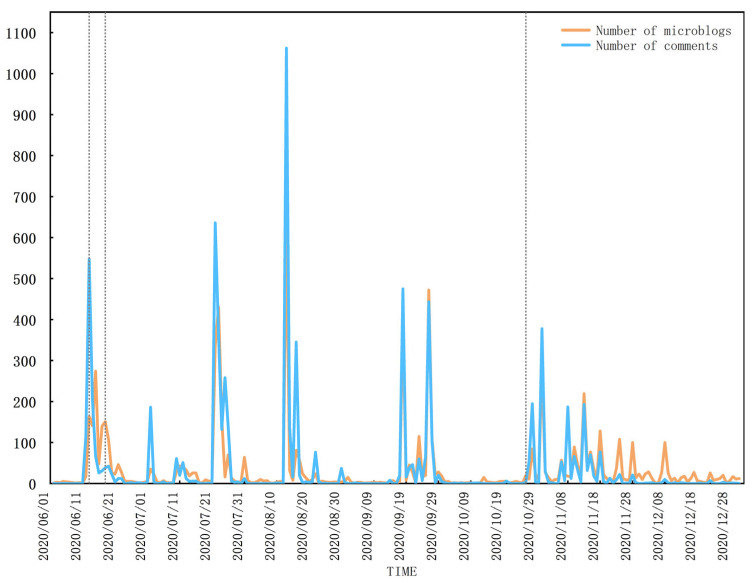
Life-cycle stage of online public opinion on imported cold-chain aquatic product safety.

**Figure 2 foods-12-01033-f002:**
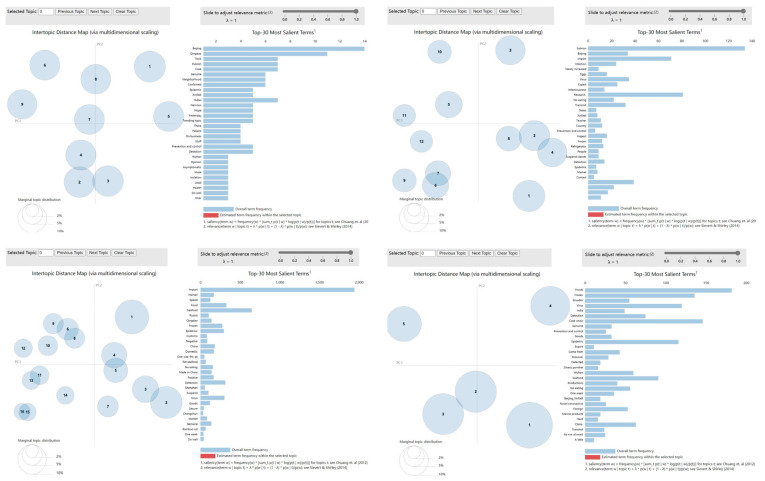
Visualization results of the topic clustering of netizen comments at each stage of public opinion.

**Table 1 foods-12-01033-t001:** Key events and heat values at each stage of public opinion.

Stage	Weibo Text Topic	Release Date	User Name	Heat Value *
I	Another case of COVID-19 was confirmed after 56 consecutive days of no additions in Beijing	12 June 2020	Capital Health	1855.97
II	COVID-19 was detected on the cutting board of imported salmon, and Beijing Xinfadi wholesale market was temporarily closed	13 June 2020	National Business Daily	1931.09
	Experts said that salmon contaminated by the virus did not represent the infection of COVID-19		Fan_Original Nutritional Information	4761.805
	Customs and other parties responded why quarantine of imported aquatic products temporarily did not include COVID-19		China Business Network	2607
	All provinces and cities investigated farm markets, supermarkets and wholesale markets, and urgently removed salmon and other imported aquatic products from the market	14 June 2020	Sina Hunan	718.805
III	Customs carried out COVID-19 risk monitoring for imported goods such as fresh cold chain products from high-risk countries or regions	18 June 2020	China News	45.485
			Healthy China	146.41
	Experts pointed out that aquatic products could be contaminated with COVID-19 rather than infected, and there was no evidence that fish can be infected with COVID-19	2 July 2020	CCTV News	1210.345
	COVID-19 was detected in the outer packaging of imported aquatic products	10 July 2020	Issued by Customs	67.89
	Business staff related to imported aquatic products was confirmed positive	22 July 2020	CCTV News	12,553.605
	The surface of imported frozen meat products tested positive for COVID-19	13 August 2020	CCTV News	4743.965
	The survey of the epidemic in Dalian was basically completed. It was speculated that a variety of imported aquatic products and outer packaging continued to be polluted, and the environment was polluted after the staff was exposed to infection	22 August 2020	CCTV News	385.035
	COVID-19 was detected in imported aquatic products from many countries	23 September 2020	People Network	1152.19
IV	Conclusion: The virus in Beijing Xifadi came from imported cold chain food	27 October 2020	Beijing Evening News	2833.28

* Combined with related literature, the rule of calculating the heat value of microblogs in this paper is defined as follows: heat value = 0.2 × number of likes + 0.365 × number of retweets + 0.435 × number of comments.

**Table 2 foods-12-01033-t002:** Topic words and a summary of the content that netizens focus on at each stage of public opinion.

Stage	Number	Topic Words	Content	Number	Topic Words	Content
	Topic 1–1	Case, confirmed, Wuhan, infection source	Virus source	Topic 1–6	Neighborhood, Xinfadi, place, detail	Publish the patient’s origin
	Topic 1–2	Beijing, protection, activity, fast speed	Praise the efficiency of epidemic prevention	Topic 1–7	Track, Announcement, detection, on business	Hope to publish the track
I	Topic 1–3	Qingdao, Hubei, newly increased, contact history	Fear of the spread of the epidemic	Topic 1–8	Degree, have reached, infection, condition	Consult the patient’s symptoms
	Topic 1–4	Hubei, quarantine, infected area, reappearance	Consult isolation measures	Topic 1–9	Time, Fengtai, capital, report, suggestion	Condemn the efficiency of information disclosure
	Topic 1–5	Announcement, top search, pandemic, condition	Question the control of comments			
	Topic 2–1	Monitor, speed, upgrade, hurry	Condemn the efficiency of information disclosure	Topic 2–7	Aquatic products, import, detected, COVID-19	Fear of the safety of imported aquatic products
	Topic 2–2	Xinfadi, case, detail, track	Condemn that a detailed trail of action is not published	Topic 2–8	Expert, announcement, research, risk	Question the opinion of experts
II	Topic 2–3	Salmon, many, investigation, direction	Ask about the supply and whereabouts of salmon	Topic 2–9	Import, detection, product, department, work	Propose testing the relevant staff rather than goods
	Topic 2–4	Salmon, source, bat, virus	Suspect the blame on salmon	Topic 2–10	Newly increased, six cases, mask, key, no wearing	Condemn no wearing of a mask
	Topic 2–5	Teacher, contact, detection, hand-washing, fish meat	Proper food processing methods	Topic 2–11	Detection, at ease, speed, prevention and control	Praise the response speed
	Topic 2–6	Virus, contaminate, seafood, customs, inspection	Question the lack of virus testing when imported	Topic 2–12	Salmon, environment, source, virus	Fear salmon carrying the virus
	Topic 3–1	Virus, Wuhan, Beijing, seafood market, source	Virus source	Topic 3–9	Import, suspend, ban, danger, cold storage	Support the suspension of imports
	Topic 3–2	Salmon, out of market, Dalian, Qingdao	Remove salmon from the market everywhere	Topic 3–10	Process, imported aquatic products, work, enterprise	Virus contamination link
	Topic 3–3	Detection, infection, contaminate, COVID-19	Salmon may be polluted	Topic 3–11	Food, relaxation, customs, nationwide, infection	Condemn the lack of food safety supervision and management
	Topic 3–4	Ask, import, food, disinfection, cold storage	Proper food processing methods	Topic 3–12	On earth, infection, pollution, terror, key	Condemn the experts for being vague
III	Topic 3–5	Import, customs, detection, meat, negative	Customs announced the test results of imported products	Topic 3–13	Dalian, Liaoning, Kaiyang, confirmed, Qingdao, detected	Relevant staff carried COVID-19
	Topic 3–6	Xi’an, staff, epidemic prevention, work hard, detail	Encourage epidemic prevention personnel	Topic 3–14	Danger, less eating, eating seafood, transmit	Fear aquatic products infected with the virus
	Topic 3–7	Pandemic, bamboo rat, one-size-fits-all, work	Condemn one-size-fits-all bamboo rat farming	Topic 3–15	Import, no buying, no eating, aquatic products	No buying of imported aquatic products
	Topic 3–8	No eating, aquatic products, fear, pandemic	No eating aquatic products	Topic 3–16	Belt fish, chicken wing, positive, Brazil, surface, Indonesia	Imported frozen products were detected with COVID-19
	Topic 4–1	Seafood, detection, Wuhan, source, import, outbreak	Link to the epidemic of the Huanan seafood market in Wuhan	Topic 4–4	Aquatic products, made in China, China	Support made in China
IV	Topic 4–2	Import, foreign, no way, ban, stop	Question why it was imported	Topic 4–5	India, one week, suspend, no way, strict prevention	Insufficient efforts for prevention and control of import countries
	Topic 4–3	Ecuador, India, product, positive	COVID-19 was detected in foreign aquatic products			

**Table 3 foods-12-01033-t003:** Emotion classification and emotion value at each stage of public opinion.

	Positive Comments			Neutral Comments			Negative Comments		
Stage	Number	Ratio	Mean	Number	Ratio	Mean	Number	Ratio	Mean
I	36	31%	0.8187	5	4%	0.5098	74	64%	0.0683
II	221	25%	0.8515	35	4%	0.5005	620	71%	0.0662
III	1610	32%	0.8680	172	3%	0.4977	3315	65%	0.0839
IV	387	25%	0.8350	52	3%	0.5071	1104	72%	0.0744
Total	2254	30%	0.8599	264	3%	0.5002	5113	67%	0.0795

## Data Availability

The data presented in this study are available on request from the corresponding author.

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
