# Peer review of "Crisis Management Experience from Social Media: Public Response to the Safety Crisis of Imported Aquatic Products in China during the Pandemic"

_foods, 2023, doi:10.3390/foods12051033_

Round 1

Reviewer 1 Report

The paper examines an interesting topic, but the novelty of the subject is exclusively related to the provision of data from a specific event in China in 2020, and the findings of the study in my opinion, it is difficult to generalize due to the limited impact of this case study. The study examines the evolution in 2020 of only one type of imported cold chain aquatic product (salmon) during the pandemic and with limited safety implications (????). On the other hand, online Chinese public opinion participated at certain times (mostly after media coverage). These facts give a limited scope in my opinion to this particular case study, additionally, it does not present the object of study in comparison to similar cases internationally.

1. In the Abstract, which is quite detailed in terms of results and proposals, the purpose of the research is not included. Also, the study is a classic case study where, however, the Summary does not make it clear which specific cold chain safety incident of imported aquatic product it refers to and in which country (this lack is also observed in the title, which we propose to change in more specific and less general).

2. The reader should go to subsection 2.1 (research subject and data) to understand what specific case the paper refers to. The Introduction section remains in general consideration of the subject as it is presented in the title (regardless of whether we are dealing with a case study on the substance), without it specifying, even when presenting the aims of the study.

3. It would also be good to give a short structure of the manuscript at the end of the Introduction section.

4.  Please present more information about Sina Weibo, and present also the selection criteria for the exclusive use of this social media.

5. In my opinion, it is necessary to add a Discussion section. There is no substantial discussion of any result and there is no comparison of findings with results of similar studies internationally oriented.

6. In the Conclusion section there are mainly repetitions that we met in the previous sections. Please avoid repetitions as in the Conclusion section the authors have the chance to present generalized conclusions which are simply based on the results. 

7. There are no research limitations and future research.

8. The authors do not present the contribution of the paper and its originality.

9. References are limited and need to be enriched. Important references are missing.

10. There are too many repetitions in the whole manuscript. These repetitions add nothing and make the manuscript less clear.

Author Response

Dear reviewer:

Thank you for your letter and comments on our manuscript entitled "Crisis Management Experience from Social Media: Public Response to the Safety Crisis of Imported Aquatic Products During the Pandemic" (ID: foods-2181596). Those comments are very helpful for revising and improving our paper. We have studied the comments carefully and made corrections which we hope meet with approval.

We uploaded our point-by-point response to the comments. Please see the attachment.

We would like to express our great appreciation to you for comments on our paper.

Thank you and best regards.

Ms. Liu

Reviewer 2 Report

Dear authors, I read your manuscript and I found it interestingly. There are suggestions in the attached file, but above all I have two concerns:

1. The paper looks more like a research reflection than original article

2. Considering the profile of the manuscript (anthropological) perhaps the journal foods is not the most suitable 

Best regards

Author Response

(The authors gave the same response as above.)

Round 2

Reviewer 1 Report

No comments

Author Response

Dear reviewer:

Thank you very much for your comments and professional advice. Based on your suggestion and request, we have made the necessary corrections to the revised manuscript. We made a point-by-point response to the comments and uploaded a marked manuscript with changes indicated and a clean updated manuscript without highlights. Please see the attachments.

Thank you and best regards,

Ms. Liu

Reviewer 2 Report

Dear authors,

Actually the paper is ok, however (as indicated in the file), a larger reasoning and better bibliography about Python software are required

Thanks

Yours Sincerely

Author Response

(The authors gave the same response as above.)
